# UPF1: From mRNA Surveillance to Protein Quality Control

**DOI:** 10.3390/biomedicines9080995

**Published:** 2021-08-11

**Authors:** Hyun Jung Hwang, Yeonkyoung Park, Yoon Ki Kim

**Affiliations:** 1Creative Research Initiatives Center for Molecular Biology of Translation, Korea University, Seoul 02841, Korea; koukentei@hanmail.net (H.J.H.); lovoap@korea.ac.kr (Y.P.); 2Division of Life Sciences, Korea University, Seoul 02841, Korea

**Keywords:** nonsense-mediated mRNA decay, UPF1, aggresome, CTIF, mRNA surveillance, protein quality control

## Abstract

Selective recognition and removal of faulty transcripts and misfolded polypeptides are crucial for cell viability. In eukaryotic cells, nonsense-mediated mRNA decay (NMD) constitutes an mRNA surveillance pathway for sensing and degrading aberrant transcripts harboring premature termination codons (PTCs). NMD functions also as a post-transcriptional gene regulatory mechanism by downregulating naturally occurring mRNAs. As NMD is activated only after a ribosome reaches a PTC, PTC-containing mRNAs inevitably produce truncated and potentially misfolded polypeptides as byproducts. To cope with the emergence of misfolded polypeptides, eukaryotic cells have evolved sophisticated mechanisms such as chaperone-mediated protein refolding, rapid degradation of misfolded polypeptides through the ubiquitin–proteasome system, and sequestration of misfolded polypeptides to the aggresome for autophagy-mediated degradation. In this review, we discuss how UPF1, a key NMD factor, contributes to the selective removal of faulty transcripts via NMD at the molecular level. We then highlight recent advances on UPF1-mediated communication between mRNA surveillance and protein quality control.

## 1. Introduction

### 1.1. Principles of Nonsense-Mediated mRNA Decay

Ensuring the fidelity of genetic information is crucial for cell survival. In particular, the quality and quantity of mRNAs should be tightly managed because mRNAs transcribed from DNA are used as templates for protein synthesis [1]. To achieve this, eukaryotic cells have evolved highly sophisticated mRNA quality control mechanisms typified by nonsense-mediated mRNA decay (NMD) [1,2,3]. In general, the presence of premature termination codons (PTCs) within an open reading frame (ORF) tends to reduce the half-life of mRNAs through the NMD pathway. NMD is associated with various pathological and physiological events as well as with variations in the clinical severity of such events [4]. For instance, it is estimated that approximately one-third of human genetic diseases are thought to be caused by PTCs on mRNAs; consequently, their occurrence can be regulated by NMD [5,6]. In addition, depending on the type of PTC-containing mRNA and tumor microenvironment, cancer cells take advantage of the pro- or anti-tumorigenic abilities of NMD [7].

It should be noted that, in addition to the faulty mRNAs containing PTCs, NMD can target naturally occurring normal mRNAs by acting as a post-transcriptional gene-regulatory mechanism [2]. These endogenous natural mRNAs targeted for NMD include mRNAs harboring upstream ORFs in the 5’-untranslated regions (5’UTRs), a long 3’UTR, exon junction complexes (EJCs) at the 3’UTR, or the UGA encoding selenocysteine within the ORF [2]. Such post-transcriptional regulation via NMD participates in successful adaptation to various intrinsic or extrinsic stresses as well as many cellular and biological processes, such as cell differentiation, cell death, viral defense, and organismal development [7,8,9].

### 1.2. NMD Occurs during Translation

Pre-mRNAs newly synthesized by RNA polymerase II in the nucleus are subjected to multiple processes, including 5’-capping, 3’-polyadenylation, and splicing [10]. During these processes, the 5’-cap structure of pre-mRNAs is bound by the nuclear cap-binding complex (CBC), a heterodimer composed of cap-binding proteins 80 and 20 [11,12,13,14]. During or after these steps, properly processed mature mRNAs with their 5’-caps bound by CBC are exported to the cytoplasm via nuclear pores. The mRNAs that are in the process of being exported or have already been exported are subject to two sequential translation events. The CBC recruits the 40S ribosomal subunit (a small subunit of the ribosome) of the ribosome with the help of the CBC-dependent translation initiation factor (CTIF), which acts as a molecular scaffold located on the cytoplasmic side of the nuclear envelope [11,12,15]. The recruited 40S ribosomal subunit scans the 5’UTR and initiates protein synthesis at the AUG codon in a proper context after joining with the 60S ribosomal subunit (a large subunit of the ribosome). As this modality occurs during the first translation of newly synthesized mRNAs, it is referred to as “the pioneer round of translation” (pioneer translation, also called the first round of translation) [16]. It should be noted that pioneer translation ensures mainly mRNA quality control rather than massive protein synthesis.

Messenger ribonucleoproteins associated with CBC are drastically remodeled during or after their export [11,12,13,14,17,18]. One peculiar event is the replacement of CBC by eukaryotic translation initiation factor 4E (eIF4E), the major cytoplasmic cap-binding protein, in a translation-independent manner [17,18,19,20,21]. eIF4E plays a crucial role during conventional protein synthesis by exploiting another scaffold protein, eIF4G, which directly associates with the eIF3 complex and recruits the 40S ribosomal subunit. Unlike the pioneer translation mediated by CBC, eIF4E actively drives multiple rounds of translation, generating a large amount of proteins. Hence, eIF4E-driven translation is referred to as steady-state translation [22,23,24].

To efficiently remove PTC-containing mRNAs, the NMD machinery should be able to distinguish PTC-containing mRNAs from normal mRNAs. Successful discrimination by the NMD machinery necessitates a translation event to determine whether a given termination codon is normal or premature [2,3]. In general, NMD is coupled to the pioneer translation [2,3]. However, several studies have revealed that a steady-state translation also contributes to PTC recognition during NMD [25,26]. Nonetheless, it remains unclear what factors determine the coupling of NMD to the pioneer translation, steady-state translation, or both.

### 1.3. Molecular Mechanism Underlying NMD

It is well known that the canonical NMD pathway is dependent on EJCs and translation events [2,3,6,7,27]. During pre-mRNA splicing in the nucleus, EJCs are deposited onto mRNAs ~20–24 nucleotides upstream of each exon–exon junction. EJCs loaded onto spliced mRNAs play multiple roles in post-transcriptional gene regulation, including splicing, mRNA export, translation, and NMD [28,29,30]. In addition, recent investigations have revealed that EJCs increase mRNA fidelity by preventing over-splicing or re-splicing events of mature mRNAs [31,32]. As most translation termination codons are located in the last exon, EJCs are efficiently removed by translating ribosomes. However, the presence of PTCs ~50–55 nucleotides upstream of the last exon–exon junction causes premature translation termination and leaves EJCs downstream of PTCs. Failure to dissociate EJCs from mRNAs leads to rapid degradation of mRNA by the NMD pathway.

When a ribosome encounters a normal termination codon, the eukaryotic peptide chain release factor 1/3 (eRF1/3) complex enters the ribosome A site, facilitates the release of nascent polypeptides for the translation termination complex, and helps to efficiently recycle the ribosomes [33,34]. In contrast, when mRNAs contain PTCs, the eRF1/3 complex recruits up-frameshift 1 (UPF1) and suppressor with morphogenetic effect on genitalia 1 (SMG1), leading to the formation of the SMG1-UPF1-eRF1/3 (SURF) complex (Figure 1) [35]. Further interaction between the SURF complex and an EJC downstream of PTC activates SMG1 kinase to induce hyperphosphorylation of UPF1 (Figure 2). Hyperphosphorylated UPF1 inhibits additional rounds of translation [36] and recruits RNA-degrading enzymes such as SMG6 as well as other adapter proteins, including SMG5, SMG7, and proline-rich nuclear receptor coactivator 2 (PNRC2), to trigger the rapid degradation of mRNA [3]. SMG5 and PNRC2 recruit a decapping complex composed of DCP1A and DCP2 to elicit decapping followed by 5’-to-3’ mRNA degradation [37,38]. SMG5 also interacts with SMG7 and recruits the CCR4-NOT complex to elicit deadenylation followed by 3’-to-5’ mRNA degradation [39]. SMG6 has endoribonucleolytic activity and cleaves internally in the vicinity of the PTC [40].

In addition to canonical NMD, which depends on EJCs, noncanonical NMD acts on a subset of mRNAs. For instance, mRNAs with a long 3’UTR (over 1000 nucleotides in length) are preferentially targeted for NMD [41]. According to a widely accepted model, the cytoplasmic poly(A)-binding protein PABPC1, which associates with the poly(A) tail of mRNAs, promotes efficient translation termination. However, in the case of mRNAs with a long 3’UTR, the interaction between PABPC1 and the terminating ribosome complex is inefficient. Consequently, the terminating ribosome complex preferentially recruits NMD factors, leading to the rapid degradation of mRNAs. Although such noncanonical NMD is initiated in an EJC-independent manner, it also exploits canonical NMD factors for efficient mRNA degradation [39,42,43].

## 2. UPF1-Mediated Degradation of Truncated Polypeptides Generated from PTC-Containing mRNAs

### 2.1. Truncated Polypeptides Are Generated from PTC-Containing mRNAs

As mentioned above, efficient NMD requires at least a single round of translation to monitor the position of PTC on mRNAs. However, it is important to stress that, in general, NMD does not completely eliminate its substrates from cells, but lowers the level of PTC-containing mRNAs to approximately 5–25% of that corresponding to PTC-free mRNAs [44]. Recent transcriptome-wide analyses also show that approximately 50% of naturally occurring mRNA variants harboring PTCs can escape NMD with different efficiencies [4,45,46]. Furthermore, real-time visualization of single PTC-containing mRNA molecules in living cells has revealed that approximately 5–30% of them escape NMD [47]. Another recent study using SunTag translation imaging revealed that the cleavage rate of PTC-containing mRNA molecules remains constant for over 100 rounds of translation [48]. These observations indicate that, despite being destined for NMD, PTC-containing mRNAs can express significant amounts of truncated polypeptides within cells. Given that (i) truncated polypeptides could be potentially misfolded, and (ii) accumulation of misfolded polypeptides induces proteotoxic stress, the truncated polypeptides generated from PTC-containing mRNAs should be selectively recognized and removed from cells.

### 2.2. Role of UPF1 in the Rapid Degradation of Truncated Polypeptides Generated from PTC-Containing mRNAs

To investigate the fate of truncated polypeptides generated from PTC-containing mRNAs (PTC-polypeptides), Kuroha et al. employed reporter mRNAs harboring a PTC at different positions and measured the relative abundance of PTC-containing mRNAs and PTC-polypeptides in yeast [49]. They observed that the yeast Upf1 contributes to a reduction in the amount of PTC-polypeptides and PTC-containing mRNAs [49]. Intriguingly, they also found that the observed reduction in the level of PTC-polypeptides by Upf1 requires a proteasome and sufficiently long 3’UTR [49,50], suggesting a possible interplay between the rapid degradation of PTC-containing mRNA via NMD and proteasomal degradation of PTC-polypeptides via Upf1. A subsequent study revealed an association between a PTC-polypeptide and heat shock protein 70 (Hsp70) in yeast [50]. Yeast Upf1 (I) associates with Sse1, a component of the Hsp90 complex comprising Hsp70, Sti1, and Sse1 [51]; (II) possesses E3 ubiquitin ligase ability (Figure 2) [52]; (III) is recruited to a terminating ribosome complex on PTC along with eRF1/3 during NMD. Accordingly, Upf1 recruited to the PTC-containing mRNAs during NMD may recognize PTC-polypeptides via the Hsp90 complex. Concomitantly, the E3 ligase activity of Upf1 may drive proteasomal degradation of these PTC-polypeptides. Thus, UPF1 may play a role in both mRNA surveillance and protein quality control (Figure 3A).

## 3. Role of UPF1 in the Ubiquitin–Proteasome System-Mediated Degradation of Proteins

### 3.1. The Ubiquitin–Proteasome System

During translation, nascent polypeptides are properly folded into their native conformations with the help of chaperones [53]. For instance, HSP70 binds to incompletely folded proteins, guiding them to be correctly folded, refolded, or disaggregated in an ATP-dependent manner. However, prematurely terminated polypeptides can still be synthesized because of the inevitable imperfections inherent to the translation machinery. These defective ribosomal products may also be potentially misfolded or aggregated. In addition to the defective ribosomal products, properly folded proteins can be converted into misfolded or aggregated proteins under stressful conditions, such as exposure to reactive oxygen species, ultraviolet damage, osmotic stress, and thermal stress [54,55]. Accumulation of such misfolded proteins within cells is deleterious to normal cell function and viability. Therefore, to cope with the accumulated misfolded polypeptides, cells exploit several protein quality control pathways, typified by the ubiquitin–proteasome system (UPS), which is responsible for the degradation of over 80% of intracellular proteins [56,57,58,59,60].

Ubiquitin protein is a small protein (~8.5 kDa), which becomes covalently conjugated with a substrate through its C-terminal glycine residue. Although the molecular details of ubiquitin conjugation reactions vary according to the type of ubiquitin and ubiquitin-like proteins, all conjugation reactions are catalyzed by the following three key enzymes [60,61,62]: E1 ubiquitin-activating enzyme, E2 ubiquitin-conjugating enzyme, and E3 ubiquitin ligase. First, the E1 enzyme catalyzes ubiquitin C-terminal acyl adenylation and transfers activated ubiquitin to E2. The E2 enzyme is complexed with an E3 ligase associated with a target protein. The E2 enzyme transfers ubiquitin to this target protein recognized by the E3 ligase. Ubiquitin is often conjugated to target proteins as a polymer. In this case, additional ubiquitin molecules are conjugated with one of several lysine residues in the ubiquitin molecule previously conjugated with the target proteins. K48- and K63-linked polyubiquitin chains are the most well-characterized polyubiquitin chains [61,62]. The K48-linked polyubiquitin chain delivers target substrates to the proteasome, where ubiquitinated target substrates are degraded in an ATP-dependent manner [61,62]. In contrast, the K63-linked polyubiquitin chain is largely involved in non-proteolytic functions, such as DNA damage repair, cellular signaling, aggresome formation, and ribosome biogenesis [63,64,65,66].

### 3.2. UPF1 as an E3 Ligase

The N-terminal cysteine- and histidine-rich domain (CH domain) of yeast Upf1 is structurally similar to the RING-box or U-box found in E3 ligase (Figure 2) [67]. A subsequent study revealed that the CH domain of the yeast Upf1 interacts with yeast E2 Ubc3 and plays a role in in vitro self-ubiquitination [52]. This intrinsic E3 ligase ability of Upf1 promotes its association with Upf3 and improves NMD efficiency [52]. However, it remains unknown whether the E3 ligase ability of Upf1 targets any additional proteins during NMD. Nonetheless, another study has clearly shown that the E3 ligase ability of mammalian UPF1 is involved in the rapid degradation of cellular proteins unrelated to NMD. Feng et al. (2017) showed that the transcription factor MYOD protein, a master regulator of myogenesis, is stabilized upon UPF1 downregulation [68]. Interestingly, *MYOD* mRNA was not affected by UPF1 downregulation, suggesting a potential role of UPF1 in protein degradation. Furthermore, the authors demonstrated that UPF1 functions as an E3 ligase via its CH domain, promoting ubiquitination and degradation of MYOD protein, with consequent effects on myogenesis [68]. Thus, UPF1, a key NMD factor, contributes to the rapid degradation of normally occurring, well-folded, and functionally active proteins, as well as abnormal PTC-polypeptides (Figure 3B).

## 4. UPF1-Mediated Aggresome Formation

### 4.1. The Aggresome

When the functions of the UPS are impaired due to internal or external damage or when the expression of misfolded polypeptides exceeds the UPS capacity, the misfolded polypeptides tend to form small cytoplasmic aggregates [65]. These aggregates are transported in a retrograde manner toward the minus end of the microtubule-organizing center via the dynein motor protein. In this way, small aggregates accumulate around the nucleus and form a large non-membranous cellular compartment, the so-called aggresome, which is then surrounded by vimentin and is ultimately degraded by the autophagy pathway [69,70,71].

Efficient sequestration of misfolded polypeptides in an aggresome requires the selective recognition of misfolded polypeptides through several adaptor proteins, such as histone deacetylase 6 (HDAC6), BAG cochaperone 3 (BAG3), as well as the CED complex composed of CTIF, eukaryotic translation elongation factor 1 alpha 1 (eEF1A1), and dynactin 1 (DCTN1). HDAC6 binds to both dynein motor protein and polyubiquitinated misfolded proteins and enhances the formation of an aggresome containing polyubiquitinated misfolded proteins [72]. BAG3 contains the PxxP and BAG domains, which interact with dynein and HSP70, respectively, stimulating aggresomal targeting of both polyubiquitinated and non-polyubiquitinated polypeptides [73,74]. The CED complex is engaged in the polyubiquitination-dependent formation of an aggresome containing misfolded polypeptides [75,76,77]. Originally, eEF1A1, a component of the CED complex, was characterized as a translation elongation factor that interacts with aminoacyl-tRNA to deliver it to the A site of ribosomes in a GTP-dependent manner [78]. Interestingly, eEF1A1 is also known to associate with misfolded polypeptides and generate a signal for aggresome formation [79,80,81]. Another CED component, DCTN1, is the largest subunit of the dynactin complex, which interacts with dynein motor protein and microtubules [82]. Within the CED complex, CTIF acts as a scaffold protein linking eEF1A1-misfolded polypeptides with DCTN1-dynein motor protein [75]. This allows the entire complex consisting of misfolded polypeptides, eEF1A1, CTIF, DCTN1, and dynein motor protein, to be transported to the aggresome via retrograde movement along microtubules. Of note, it is known that the CED complex functions in concert with HDAC6 to form an aggresome containing a certain type of misfolded polypeptide [75]; however, the parameters determining the choice of specific adaptor proteins remain unknown.

Aggresomes are closely associated with neurodegenerative diseases. Aggresomes observed in cultured cells are generally considered to be morphologically and biochemically similar to cytoplasmic inclusion bodies observed in various neurodegenerative diseases [83], such as Lewy bodies in Parkinson’s disease [72], glutamine repeats (polyQ aggregates) in Huntington’s disease [84], mutant superoxide dismutase 1 in Lou Gehrig’s disease [73], and Tau aggregates in Alzheimer’s disease [85,86]. Therefore, an in-depth understanding of the molecular mechanisms underlying the formation and removal of aggresomes will provide promising therapeutic approaches for the treatment of neurodegenerative diseases.

In addition to neurodegenerative diseases, aggresomes or aggresome-like structures are involved in the efficient replication or packing of several DNA or RNA viruses, such as influenza A viruses, adenovirus, herpes simplex virus, and African swine fever virus [87,88,89,90,91,92,93,94,95]. A recent study also pointed to the mechanistic similarity between inflammasome assembly and aggresome formation, suggesting that aggresome and aggresomal targeting proteins (HDAC6, BAG3, and CED complex) could play a role in inflammasome formation and regulation [96].

### 4.2. UPF1 as an Aggresomal Targeting Factor

As mentioned above, when the UPS is impaired or overwhelmed, misfolded polypeptides tend to accumulate in aggresomes. As PTC-containing mRNAs targeted for NMD always generate truncated nascent polypeptides (PTC-polypeptides) as byproducts [57,97], such potentially misfolded polypeptides can be transported toward the aggresome upon UPS impairment. Indeed, a recent study showed that PTC-polypeptides generated from a reporter NMD substrate (glutathione peroxidase 1 mRNA harboring a PTC) are enriched in aggresomes after treating cells with MG132, a potent proteasome inhibitor [76]. This observation suggested a possible crosstalk between the NMD pathway and aggresome formation. Further analysis revealed that efficient formation of aggresomes containing PTC-polypeptides requires CED components and UPF1, but not other canonical NMD factors, such as PNRC2, SMG5, SMG6, and SMG7. This finding highlighted a specific role of UPF1 in aggresome formation. Interestingly, UPF1-mediated aggresome formation is not limited to PTC-polypeptides. In addition to PTC-polypeptides, UPF1 contributes to the efficient formation of aggresomes containing other misfolded polypeptides that are irrelevant to PTC; these include (i) a mutant form of cystic fibrosis transmembrane conductance regulator harboring a single amino acid deletion at position 508 (CFTR-ΔF508) [69,72,75] and (ii) puromycin-conjugated polypeptides generated upon puromycin treatment, which causes premature translation termination and releases truncated polypeptides in puromycin-conjugated forms [98]. This indicates that UPF1 plays a dual role in protein quality control by modulating aggresome formation, and in mRNA quality control via NMD (Figure 3C).

Various studies have attempted to understand the molecular mechanism underlying UPF1-mediated aggresome formation. Based on the fact that efficient NMD is dependent on a continuous cycle of phosphorylation and dephosphorylation of UPF1, Park et al. (2020) investigated the possible role of UPF1 phosphorylation in aggresome formation [76]. First, they noted that aggresomal targeting of UPF1 was dependent on its phosphorylation status. Second, hyperphosphorylated UPF1 associated preferentially with the CED complex and misfolded polypeptides, stabilizing the complex. Intriguingly, this preferential association was not affected by a mutation that renders hyperphosphorylated UPF1 to lack E3 ligase ability, indicating that the E3 ligase ability of UPF1 is dispensable for aggresome formation. Third, visualization and single-particle tracking of CTIF aggregates in live cells using line-scan confocal microscopy showed that downregulation of UPF1 inhibited active movement of CTIF aggregates toward aggresomes. Taken together, these observations suggest that UPF1 facilitates efficient aggresome formation by stabilizing the CED complex associated with misfolded polypeptides and by ensuring proper movement of this complex to the aggresome.

## 5. Conclusions

It has long been considered that UPF1 is a specific factor responsible for mRNA surveillance and other mRNA decay pathways [3]. In addition to this function, UPF1 contributes to protein quality control in several different ways: first, UPF1 specifically degrades PTC-polypeptides in an NMD-coupled manner [49,50]. Second, UPF1 acts as an E3 ligase to degrade target proteins in an NMD-independent manner [68]. Third, UPF1 senses the CED complex associated with misfolded polypeptides, and guides it to move toward the aggresome in an NMD-independent manner [76]. Thus, UPF1 ensures proper gene expression and protects the cells against the accumulation of misfolded polypeptides (Figure 3).

It should be noted that, in addition to sequestration of misfolded polypeptides, aggresomes and aggresome-like structures are associated with several other processes, such as inflammasome formation [96] and efficient replication, proliferation, and packing of viruses [87,88,89,90,91,92,93,94,95]. Given that UPF1 functions as an enhancer of aggresome formation, the above observations implicate that UPF1 could regulate a variety of cellular and biological processes, more than we appreciated before.

## Figures and Tables

**Figure 1 biomedicines-09-00995-f001:**
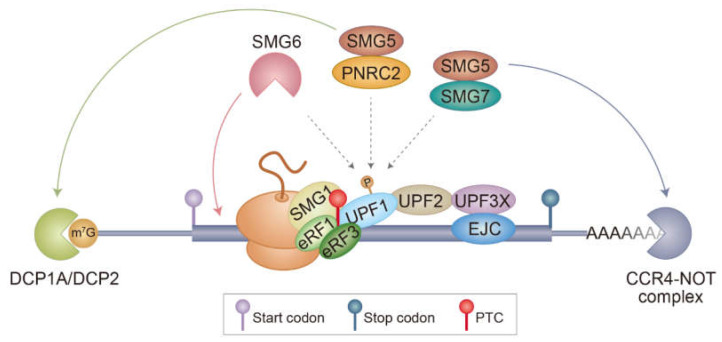
A model for canonical NMD. PTC is recognized during translation mediated by either CBC or eIF4E. A terminating ribosome on a PTC recruits SMG1 kinase, UPF1, and eRF1/3, forming the SURF complex. Association between the SURF complex and a PTC-distal EJC activates SMG1 kinase and triggers UPF1 hyperphosphorylation. The hyperphosphorylated UPF1 recruits several adaptors or ribonucleases, such as SMG5–SMG7 and PNRC2. Eventually, PTC-containing mRNAs are subjected to (I) endoribonucleolytic cleavage by SMG6, (II) decapping followed by 5’-to-3’ exoribonucleolytic degradation by PNRC2 and SMG5, and (III) deadenylation followed by 3’-to-5’ degradation by SMG5 and SMG7. Notably, the rapid degradation of NMD substrates may require all three mRNA decay pathways to act simultaneously or, alternatively, through one or two predominant pathways. The detailed molecular mechanisms underlying the choice between the above options need to be addressed by future studies.

**Figure 2 biomedicines-09-00995-f002:**
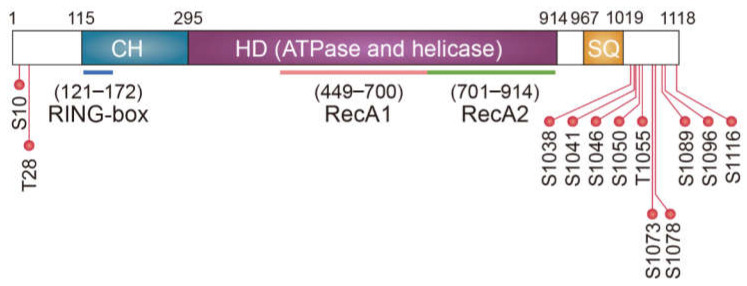
Schematic diagram of human UPF1. Specific regions and domains within UPF1 are indicated. The helicase domain is composed of two RecA-like domains. Numbers and dots indicate the position of amino acid residues and experimentally validated phosphorylation sites, respectively. CH, cysteine- and histidine-rich domain; HD, helicase domain; and SQ, serine- and glutamine-rich domain.

**Figure 3 biomedicines-09-00995-f003:**
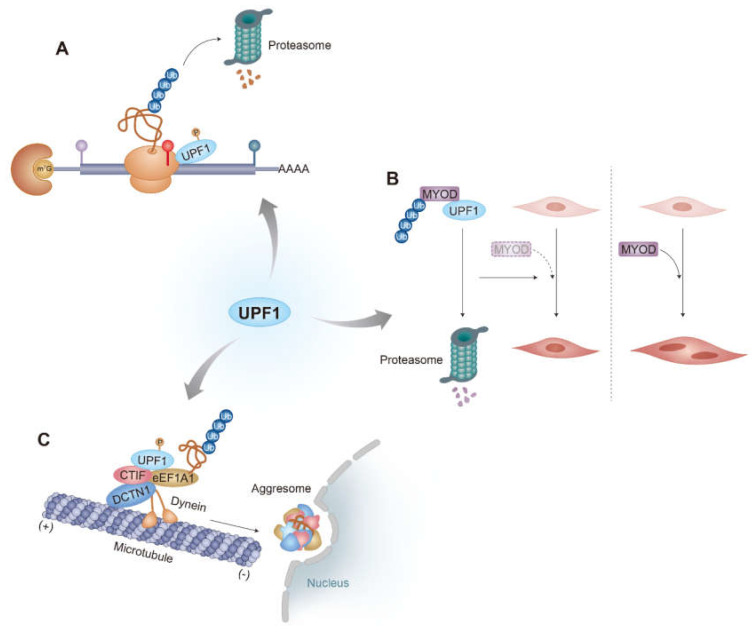
Diverse functions of UPF1 in protein quality control. (**A**) UPF1-driven degradation of PTC-polypeptides synthesized during NMD in yeast. Given its coupling to NMD, this function might be dependent on UPF1 hyperphosphorylation. The involvement of the E3 ligase ability of UPF1 and whether this mechanism works for PTC-polypeptides generated in mammalian cells remain to be determined. (**B**) UPF1-mediated degradation of properly folded proteins via the UPS. The E3 ligase activity of UPF1 is involved in ubiquitination of target proteins. A possible role of UPF1 hyperphosphorylation in this mode of protein degradation should be investigated. (**C**) UPF1-mediated degradation of misfolded polypeptides via the aggresome-autophagy pathway. UPF1 associates with the CED complex and misfolded polypeptides, ensuring their proper delivery to aggresomes. This mode of protein degradation is dependent on UPF1 hyperphosphorylation, but not on its E3 ligase activity.

## Data Availability

Not applicable.

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
