# Peer review of "UPF1: From mRNA Surveillance to Protein Quality Control"

_biomedicines, 2021, doi:10.3390/biomedicines9080995_

Round 1
Reviewer 1 Report
This is an interesting, timely and well written review article. I very much enjoyed reading it. For the most part, the subject matter is up to date and well organized, and the figures are informative. Therefore, I found little to criticize in this manuscript. I do think, however, the in the first section the authors could incorporate a bit more recent relevant information. Thus, I think it would help if they could briefly discuss and cite two recent papers:
- B. Joseph and E.C. Lai, PLOS Genetics 5/25/2021
- Omani et al. Int. J. Mol. Sci (2021) 22: 6519
Author Response
We appreciate the Reviewer’s time and effort to evaluate our work. As suggested by the Reviewer, the above two papers are now mentioned and discussed in the revised manuscript. The multiple functional roles of EJCs are also described in the revised manuscript.

Reviewer 2 Report
The manuscript introduces nonsense-mediated mRNA decay (NMD) and premature termination codons (PTCs), together with translational control processes. Next the authors introduce the NMD factor, UPF1, and its role in controlling truncated polypeptides as an E3 ubiquitin ligase through various mechanisms. This manuscript provides a comprehensive basic understanding of the aforementioned processes.
However, the scope of the journal is to publish manuscripts describing mechanistic insight and advances related to therapeutic delivery of nucleic acids to patient cells as drugs. Thus this manuscript in my opinion is outside the scope of this journal. The oly small mention of disease is in lines 273-288. In order to make the manuscript publishable, a focus needs to be included on using UPF1-related mechanisms to therapeutic use.
Another minor note is that in line 65 the information in brackets is redundant
Author Response
We are grateful for the Reviewer’s time and critical comments. This contribution is part of a series of review papers arranged by Dr. Lejeune Fabrice, the Guest editor of this Special Issue in Biomedicines. The scope of this Special Issue is to provide up-to-date molecular insight on the nonsense-mediated mRNA decay mechanism, its regulation, and its involvement in various pathologies. I was invited by the Guest editor to write a review manuscript focusing specifically on the NMD mechanism and its regulation. I believe that other experts in the field will contribute in-depth analyses of NMD-related pathologies.

Round 2
Reviewer 2 Report
If the topic was invited, then I have no problem with its inclusion